# The Assessment of the Effect of Autophagy Inhibitors—Chloroquine and 3-Methyladenine on the Antitumor Activity of Trametinib Against Amelanotic Melanoma Cells

**DOI:** 10.3390/cells14070557

**Published:** 2025-04-07

**Authors:** Dominika Stencel, Justyna Kowalska, Zuzanna Rzepka, Klaudia Banach, Marta Karkoszka-Stanowska, Dorota Wrześniok

**Affiliations:** Department of Pharmaceutical Chemistry, Faculty of Pharmaceutical Sciences in Sosnowiec, Medical University of Silesia in Katowice, 4 Jagiellońska Str., 41-200 Sosnowiec, Poland; jkowalska1902@gmail.com (J.K.); zrzepka@sum.edu.pl (Z.R.); kbanach@sum.edu.pl (K.B.); marta.karkoszka@sum.edu.pl (M.K.-S.)

**Keywords:** trametinib, chloroquine, 3-methyladenine, autophagy, apoptosis, LC3A/B, LC3B

## Abstract

Malignant melanoma, particularly amelanotic melanoma, contributes to a very serious problem in public health. One way to find new therapies is to learn about and understand the molecular pathways that regulate cancer growth and development. In the case of a tumor, the autophagy process can lead to the development or inhibition of cancer. This study aimed to assess the cytotoxicity of connection trametinib (MEK1 and MEK2 kinase inhibitor) with autophagy inhibitors—chloroquine (lysosomal clearance of autophagosomes inhibitor) and 3-methyladenine (phosphatidylinositol 3-kinases inhibitor), on two amelanotic melanoma cell lines (C32 and A-375). The results showed that combination therapy had better anti-proliferative effects than alone therapy in both cell lines. The C32 cell line was more sensitive to 3-methyladenine treatment (alone and in combinations), and the A375 line showed sensitivity to chloroquine and 3-methyladenine (alone and in combinations). The anti-proliferative effect was accompanied by dysregulation of the cell cycle, a decrease in the reduced thiols, the depolarization of the mitochondrial membrane and the level of p44/p42 MAPK. Both inhibitors have the ability to induce apoptosis. Differences in the level of LC3A/B and LC3B proteins between the chloroquine and the 3-methyladenine samples indicate that these drugs inhibit autophagy at different stages. The enhancement of the effect of trametinib by autophagy inhibitors suggests the possibility of combining drugs with anti-cancer potential with modulators of the autophagy process.

## 1. Introduction

Tumors are one of the most common causes of death in the world. The causes of its high mortality are late diagnoses, the type of cancer, and insufficient treatment. In addition, the incidence of these cancers is increasing. Despite the continuous development of science and medicine, current oncological therapies remain unsatisfactory. As the data show, cancers are the second leading cause of death in the USA. One of the most popular cancers is melanoma [1,2].

Melanoma is a malignant tumor that originates from melanin-producing cells called melanocytes. The primary site of this type of cancer is the skin, but it can also appear on the eyes, ears, gastrointestinal tract, leptomeninges, and sinonasal, oral and genital mucous membranes. The incidence of melanoma increases year by year, especially among the white population, and differs depending on the continent: in Australia, there are 50–60 new melanoma cases per 100,000 inhabitants, in the USA, 20–30 per 100,000, and in Europe 10–25 per 100,000. This type of cancer is responsible for 90% of the deaths associated with cutaneous tumors [3,4]. Moreover, melanoma is beginning to affect younger people. Risk factors include sun exposure, skin type with a tendency to burn, genetic predisposition, and atypical naevi. 

There are different types of melanoma; one of them is amelanotic/hypomelanotic melanoma. This type is distinguished by a low content of melanin, which often makes it difficult to identify; moreover, it shows the dysregulation of cell cycle control and apoptosis pathways. Amelanotic melanoma is more aggressive, has less surveillance, and has a higher recurrence rate than typical melanoma [4,5]. Melanoma diagnosis is based on observing the change in terms of asymmetry, marked cellularity, confluence of growth, and poor circumscription [6]. Treatment of this kind of cancer is based primarily on surgery, and when this is not sufficient, pharmacotherapy is introduced. Currently, modern pharmacological treatment is based on targeted therapy and immunotherapy. One of the targeted therapy drugs used in the treatment of melanoma is trametinib [7,8].

Trametinib is a MEK1 and MEK2 kinase inhibitor with anti-cancer activity. This drug has been approved for the treatment of metastatic or unresectable melanoma with a BRAF V600E or BRAF 600K mutation or as an adjuvant treatment [7]. In addition, trametinib inhibits RAF-mediated phosphorylation, resulting in the inhibition of ERK sailing activity. Trametinib causes decreased cell proliferation, G1 cell cycle arrest, and induces apoptosis. The drug is usually well-tolerated; however, it has side effects such as skin rash, diarrhea, fatigue, peripheral edema, nausea, and vomiting. There have been cases in which it was necessary to reduce the dose or discontinue trametinib treatment due to bothersome side effects [9,10].

Nowadays, autophagy is becoming more and more interesting in the context of understanding the mechanism and development of cancer. Autophagy (“self-eating”) is a process of the degradation and recycling of cellular molecules into lysosomes. The role of this process is to maintain intracellular homeostasis. There are three types of autophagy: macroautophagy, microautophagy, and chaperone-mediated autophagy (CMA). The differences between the different types of autophagy are based on the different transport of substrates to the interior of lysosomes [11,12]. Autophagy is a process that occurs in both physiological and pathological conditions. As the current data indicate, this process plays an important role in, e.g., neurodegenerative disorders, cardiovascular diseases, pulmonary disorders, hepatic disorders, renal diseases, reproductive dysfunctions, autoimmune disorders, and cancer [13]. The role of autophagy is not fully understood. This process can either lead to the suppression or progression of the cancer. The role of autophagy depends on biology and the surrounding microenvironment, tumor stage and its kind [14,15].

The aim of this study was to assess the effect of autophagy inhibitors on the antitumor activity of trametinib against amelanotic melanoma cells. An important aspect of the study was the inhibition of autophagy at different stages of the process. Therefore, chloroquine inhibited this process at the autophagosome completion phase, and 3-methyladenine at the nucleation phase. Understanding the mechanism of action and the impact of autophagy on the development of cancer may contribute to improving the quality and effectiveness of current cancer treatment. The introduction of an autophagy inhibitor into anti-cancer therapy may contribute to overcoming drug resistance and increasing the cytotoxic effect.

## 2. Materials and Methods

### 2.1. Chemicals and Reagents

Trametinib was obtained from Cayman Chemical, Biokom (Warsaw, Poland). Chloroquine phosphate, 3-methyladenine, Phosphate-Buffered Saline solution (PBS), Tween-20, RIPA Buffer, PVDF membranes, Bovine Serum Albumin (BSA), Tris-Buffered Saline (TBS), the antibiotics penicillin and amphotericin B were purchased from Sigma Aldrich Inc. (Taufkirchen, Germany). Dulbecco’s Modified Eagle Medium (DMEM), Trypsin/EDTA, and Fetal Bovine Serum (FBS) were acquired from Cytogen (Zgierz, Poland). ECL Western Blotting Substrate, Hoechst 33342, and Pierce™ BCA Protein Assay Kit were obtained from Thermo Fisher Scientific (Waltham, MA, USA). Neomycin was purchased from Amara (Kraków, Poland), and FITC-labeled annexin V was obtained from Biotium (Fremont, CA, USA). Via-1-Cassettes™ (contain acridine orange and DAPI fluorophores), NC-Slides™ A2 and A8, Annexin V Binding Buffer, Apoptosis Wash Buffer and the staining reagents Solution 3 (1 µg/mL DAPI, 0.1% triton X-100 in PBS), Solution 5 (400 µg/mL VitaBright48™, 500 µg/mL propidium iodide, 1.2 µg/mL acridine orange in DMSO), Solution 7 (200 µg/mL JC-1), Solution 8 (1 µg/mL DAPI in PBS), Solution 15 (500 µg/mL Hoechst 33342), Solution 16 (500 µg/mL propidium iodide) were acquired from ChemoMetec (Lillerød, Denmark). Primary rabbit monoclonal antibodies, anti-GAPDH, anti-LC3A/B, anti-LC3B, and anti-p44/p42 were obtained from Cell Signaling (Danvers, MA, USA). The remaining chemicals were acquired from POCH S.A. (Gliwice, Poland) or Sigma-Aldrich (Taufkirchen, Germany).

### 2.2. Cell Culture and Treatment

The research was conducted on melanoma cell lines C32 and A-375 (ATCC; Manassas, VA, USA). The cells were maintained in DMEM medium (Cytogen; Zgierz, Poland) supplemented with 10% fetal bovine serum, penicillin G (10,000 U/mL), neomycin (10 μg/mL), and amphotericin B (0.25 mg/mL) at 37° C and humidified 5% CO_2_ atmosphere.

Prior to the experiment, cells were pre-incubated in a growth medium for 24 h. The cells were exposed to trametinib (TRB) and two autophagy inhibitors, i.e., chloroquine (CQ) and 3-methyladenine (3-MA), using the following research models:

Model 1: A total of 24 h of incubation with autophagy inhibitor solution, followed by 48-h incubation with TRB solution.

Model 2: A total of 48 h of exposure to a combination of TRB and an autophagy inhibitor.

### 2.3. Cell Viability and Count Assay

The cells were treated in the models described in Section 2 using the following solutions: CQ (50 μM), 3-MA (5 mM), TRB (5 nM, 50 nM), CQ + TRB (50 μM + 5 nM and 50 μM + 50 nM), 3-MA + TRB (5 mM + 5 nM and 5 mM + 50 nM). Next, the cells were trypsinized, centrifuged, suspended in the culture medium, and loaded into the Via1-Cassette™, which included DAPI (staining dead cells) and acridine orange (staining the whole cell population). The analysis of cell number and cell viability was performed using a NucleoCounter^®^ NC-3000™ fluorescent imaging cytometer.

### 2.4. Cell Cycle Analysis

The cells were incubated in the following solutions: CQ (50 μM), 3-MA (5 mM), TRB (50 nM), CQ + TRB (50 μM + 50 nM), and 3-MA + TRB (5 mM + 50 nM). After treatment, cells were trypsinized and counted, and 1 × 10^6^ cells were incubated for 5 min at 37 °C with the lysis buffer containing DAPI. Then, a stabilization buffer was added to the stained cells and analysis was performed using a fluorescent imaging cytometer.

### 2.5. The Estimation of Cellular Reduced Glutathione Level

The analysis was performed with the use of VitaBright-48™ dye, which reacts with thiols to form a fluorescent product. The cells were exposed to CQ, 3-MA and TRB solutions as indicated in Section 3. The cells were then detached and counted, and 1 × 10^6^ cells were stained using Solution 5 reagent. The stained cells were analyzed using a fluorescent imaging cytometer.

### 2.6. Mitochondrial Membrane Potential

Cells were treated with CQ, 3-MA and TRB at the concentrations shown in Section 3. Subsequently, cells were trypsinized, and 1 × 10^6^ cells were stained with JC-1 and DAPI. Mitochondrial membrane potential was analyzed using a NucleoCounter^®^ NC-3000™ fluorescent imaging cytometer. The scatter plots determined the percentage of depolarized/apoptotic and polarized/healthy cells.

### 2.7. Annexin V Assay

The analysis was performed using Annexin V-fluorescein isothiocyanate to detect the presence of phosphatidylserine on the outer surface of the plasma membrane. Cells exposed to CQ, 3-MA and TRB solutions (as described in Section 3) were detached, and 3 × 10^5^ cells were stained with Hoechst 33342, Annexin V-fluorescein isothiocyanate, and propidium iodide. Analysis was conducted using a fluorescent imaging cytometer.

### 2.8. Western Blotting Analysis

Cells incubated in CQ, 3-MA, and TRB solutions indicated in Section 3 were lysed using RIPA buffer supplemented with phosphatase and protease inhibitors. Then, a Western blot analysis was performed according to the protocol described previously [16]. The following primary antibodies were used: rabbit anti-LC3A/B (1:1000), rabbit anti-LC3B (1:1000), rabbit anti-p44/42 (1:1000) and rabbit anti-GAPDH (1:1000). The analysis was performed using G: Box Chemi-XT4 Imaging System and GeneTools Software 4.0 (Syngene, Cambridge, UK). GAPDH was used as a loading control.

### 2.9. Statistical Analysis

The obtained results were subjected to statistical analysis using GraphPad Prism 7 (GraphPad Software, Inc., La Jolla, CA, USA). The data were calculated as standard deviations (SDs). The results were analyzed statistically using ANOVA as well as Dunnett’s and Tukey’s multiple comparisons test. Statistical significance was determined in the trials with a *p*-value less than 0.05.

## 3. Results

### 3.1. Influence of the Combination of Trametinib with Autophagy Inhibitors on Proliferation of Melanoma Cells

The initial assessment of the cytotoxic effect of trametinib and inhibitors of autophagy was an analysis of viability of cells and cell number. Two models of incubation were used during the study: pre-incubation with the inhibitor of autophagy and combination with the inhibitor of autophagy. The concentrations of chloroquine, 3-methyladenine, and trametinib were selected on the basis of literature data [17,18,19,20,21,22,23]. The obtained results are presented in Figure 1. In general, TRB is more effective at a concentration of 50 nM than 5 nM in the tested melanoma cells. Moreover, the combination of trametinib with an autophagy inhibitor is more potent than trametinib alone in both lines. We observed that the tested agents decreased both the viability (Figure 1A) and proliferation (Figure 1B) of melanoma cells, and the effect on proliferation was notably greater. Exposure of cells to CQ and TRB, in both the pre-incubation model and the simultaneous incubation model, resulted in a reduction in the viability of melanoma cells line C32 by 3–23% and cells line A375 by 1–22%, as well as cell numbers by 14–94%, and 39–85%, respectively. In the case of exposure to 3-MA and TRB, we also found minor differences between the research models used and observed that the viability of C32 cells was reduced by 7–22% and that of A375 cells by 2–20%, as well as the number of cells decreased by 71–94% and 60–81%, respectively. Taking into account the results obtained, only the model of simultaneous exposure to the autophagy inhibitor and TRB at a concentration of 50 nM was used in further studies.

A significant reduction in the cell numbers and their morphology in the samples can be observed in the microscopic images (Figure 2). The effect was noticed for the C32 and A-375 cell lines. It was observed that in the case of the C32 line, the greatest anti-proliferative effect occurred for 3-methyladenine, especially in combination with trametinib. In addition, cells in this sample showed expanded and non-spherical morphology. In the case of the A-375 line, the greatest effect was observed for chloroquine, particularly in combination with trametinib. The cells had a tendency toward being rounded. It is worth noting that in samples with trametinib, the A-375 cell line showed an expanded and non-spherical morphology.

### 3.2. Trametinib and Autophagy Inhibitors Disrupt the Cell Cycle of Melanoma Cell

The performed analysis showed that the tested agents disrupted the cell cycle of melanoma cells (Figure 3). An increase in the percentage of cells in the sub-G1 phase in both cell lines was observed. In the C32 cell line, the most significant increase was observed in 3-MA and 3-MA + TRB (1% for control vs. 22% for 3-MA and 27% for 3-MA + TRB). It was found that the incubation of cells line A375 with CQ + TRB and 3-MA + TRB caused the greatest changes in the percentage of cells in the sub-G1 phase (2% for control vs. 30% for CQ + TRB and 27% for 3-MA + TRB). In the case of the G1/G0 phase, we noted that TRB caused a significant increase in the percentage of cells in this phase for both cell lines (by about 10% for the C32 line and 14% for the A375 line). For the A375 line, the incubation of cells with 3-MA and 3-MA + TRB was associated with a decrease in the percentage of cells in the G1/G0 phase (75% for control vs. 62% for 3-MA and 65% for 3-MA + TRB), while exposure to CQ + TRB resulted in an increase in the percentage of cells in this phase (up to 87%). In the case of the C32 line, it was observed that CQ alone, as well as the combination of TRB and autophagy inhibitors, resulted in a lower percentage of cells in the G1/G0 phase (75% for control vs. 56% for CQ, 65% for CQ + TRB, and 67% for 3-MA + TRB). We found a decrease in the percentage of cells in the S-phase and the G2/M-phase in all samples of the C32 line. Meanwhile, in the A-375 cell line, only CQ caused an increase in the percentage of cells in the S-phase (10% for control vs. 20% for CQ), and there was a decrease in the percentage of cells in the G2/M phase in all samples.

### 3.3. Trametinib Combined with Inhibitors of Autophagy Decreases the Level of Reduced Thiols

Intracellular thiols such as glutathione are responsible for maintaining intracellular redox homeostasis. The low glutathione levels in the cell indicate a redox imbalance and the appearance of oxidative stress [16]. The results of the assessment of the effect of the studied agents on the level of reduced thiols in melanoma cells are shown in Figure 4. All samples showed an increase in the percentage of cells with low levels of reduced thiols, but there were notable differences between cell lines. For the C32 line, the highest percentage of cells with low levels of reduced thiols was observed for the 3-MA and 3-MA + TRB treatments (7% for control vs. 47% for 3-MA and 60% for 3-MA + TRB). Whereas, for the A-375 line, the greatest effect was present in cells treated with CQ + TRB, as well as 3-MA + TRB (17% for control vs. 70% for CQ + TRB and 43% for 3-MA + TRB). It is noteworthy that TRB enhanced the impact of autophagy inhibitors on the level of reduced thiols in melanoma cells line A-375.

### 3.4. Trametinib and Inhibitors of Autophagy Decrease Mitochondrial Membrane Potential in Melanoma Cells

The results shown in Figure 5 demonstrate that all studied agents affected mitochondrial potential in melanoma cells, with noticeable differences between cell lines. In the C32 line, the highest percentage of cells with depolarized mitochondria occurred for 3-MA + TRB treatment (16% for control vs. 41% for 3-MA + TRB), while in the other samples, it only increased to 23–28%. In the A375 line, it was observed that CQ and TRB combinations with autophagy inhibitors were associated with the greatest reduction in mitochondrial potential in cells (8% cells with depolarized mitochondria for control vs. 40% for CQ, 89% for CQ + TRB, and 71% for 3-MA + TRB).

### 3.5. Trametinib and Inhibitors of Autophagy-Induced Apoptosis

The ability of trametinib and inhibitors of autophagy to induce cell apoptosis was assessed by the annexin V assay. The obtained results are presented in Figure 6. Most of the studied cells were in the early stage of apoptosis. The percentage of cells in early apoptosis depended on the sample and kind of cell lines. In the C32 cell line, the highest percentage of cells in this phase vs. control (8%) was 80% for 3-MA, 68% for CQ + TBR and 86% for 3-MA + TRB. In the A-375 culture, the presence of TRB potentiated the influence of inhibitors, and it was 6% for control, 33% for CQ, 55% for 3-MA, 49% for CQ + TRB and 76% for 3-MA + TRB. The percentage of cells in the late stage of apoptosis also increased significantly. The highest percentage of cells in the late apoptosis was 32% for CQ + TRB in the A-375 cell line. It was worth noting that there were no necrotic cells in the analyzed samples.

### 3.6. Trametinib and Inhibitors of Autophagy Change the Level of Proteins Such as LC3A/B, LC3B and p44/p42

The Western blot analysis indicated that chloroquine significantly increased the levels of the LC3B and LC3A/B proteins in both tested cell lines (Figure 7). The increase occurred both for the chloroquine alone and for the combination of trametinib and chloroquine. In the C32 cell line, the increase in the LC3B and LC3A/B levels for CQ + TRB sample was higher than the increase for CQ alone, respectively, and the level of LC3B increased almost 5x for CQ and more than 8x for CQ + TRB; the level of LC3A/B increased more than 5x for CQ and more than 9x for CQ + TRB. In the A-375 cell line, the effect was different for each respective group; the level of LC3B was an approximately 12x increase for CQ and an approximately 10x increase for CQ + TRB; the level of LC3A/B increased more than 7x for CQ and then a 5x increase for CQ + TRB. It was observed that 3-methyladenine alone and in combination with trametinib also caused an increase in the levels of LC3B and LC3A/B in C32 melanoma cells, respectively: the level of LC3B was approximately 184% for 3-MA and 182% for 3-MA + TRB; the level of LC3A/B was approximately 207% for 3-MA, and 277% for 3-MA + TRB. The situation was different for the A-375 cell line, where the use of 3-methyladenine resulted in an increase in this protein for 3-MA + TRB (about 305%); the level of LC3A/B was about 58% for 3-MA and 71% for 3-MA + TRB.

The obtained results showed that the used substances generally decreased the level of p44/p42 MAP kinase in both tested cell lines (Figure 7). The effect was stronger and more significant for the C32 cell line, and it was approximately 58% for CQ, 45% for 3-MA, 62% for TRB, 44% for CQ+TRB, and 53% for 3-MA + TRB, respectively. In the A-375 cell line, no significant changes were observed.

## 4. Discussion

Melanoma is a serious public health problem. The high mortality rate, late diagnoses, increasing incidence, and the limited effectiveness of therapy have resulted in more effective treatments being sought. Understanding the molecular mechanism regulating the process of carcinogenesis is an important aspect of the search for appropriate therapy [3,6]. Recently, one of the more interesting processes in the context of cancer treatment is autophagy. It is a catabolic process leading to the controlled degradation of abnormal molecules, proteins, and cell organelles. Autophagy is dysregulated in neoplastic diseases, which can lead to both inhibition and progression of carcinogenesis. The course of autophagy in the process of carcinogenesis depends on the type and stage of cancer, the condition of the organism, and the treatment used [24].

The type of cancer is an important factor in regulating the process of autophagy. Previous studies showed that the use of the same treatment had different effects in the course of autophagy for amelanotic and melanotic melanoma. The use of tigecycline resulted in the induction of autophagy in amelanotic melanoma cells and inhibition in melanotic melanoma cells. Moreover, amelanotic melanoma cells were more resistant to treatment than melanotic melanoma cells. The obtained results allowed for the conclusion that regulation of the autophagy process may be an important factor in overcoming resistance to treatment [16].

The aim of the studies was to assess the effect of autophagy inhibitors on the antitumor activity of trametinib against amelanotic melanoma cells. For the studies, two amelanotic melanoma cell lines were selected (C32 and A-375) because this type of melanoma is more aggressive and more resistant to treatment than melanotic melanoma [5]. During the examinations, two autophagy inhibitors were used, such as chloroquine and 3-methyladenine. These substances were connected with the antitumor drug trametinib.

Chloroquine (CQ; 4-N-(7-chloroquinolin-4-yl)-1-N,1-N-diethylpentane-1,4-diamine) is a well-known antimalarial drug. This medicine also has activity against, e.g., rheumatoid arthritis, HIV infection, and the novel coronavirus SARS-CoV-2. This substance is recognized as an autophagy inhibitor and approved by the Food Drug Administration (FDA) as an autophagy inhibitor. CQ blocks the last step of the autophagy process by inhibiting the lysosomal clearance of autophagosomes (completion phase). Due to chloroquine’s ability to inhibit autophagy, this drug is attracting interest for its anti-cancer activity [25,26]. Autophagy suppression leading to chloroquine antitumor activity has been confirmed among others in the lung, breast, ovary, colorectal, and bladder. Moreover, CQ sensitizes cancer to chemotherapy, e.g., cisplatin, carmustine, and docetaxel [25].

3-methyladenine (3-MA; 6-amino-3-methylpurine) is used to inhibit autophagy and apoptosis under various conditions. This substance inhibits autophagy by inhibition of type III Phosphatidylinositol 3-kinases (PI-3K). This action causes blocking autophagosome formation (nucleation phase) [26,27]. Interestingly, 3-methyladenine can inhibit autophagy during nutrient deprivation and promote autophagy during suppressing starvation [28]. However, it has been shown that 3-MA increases the effect of substances with anti-cancer potential against breast cancer, colon cancer, and head and neck cancer [28,29,30].

As the data show, trametinib induces autophagic activity in melanoma cells. Treatment with trametinib, despite many satisfactory results, leads to the development of drug resistance. In particular when treatment is for melanoma with mutation BRAF V600 [31]. The results of the research show that the use of trametinib with inhibitors of autophagy indicates higher anti-cancer activity than the use of trametinib alone against cell lines C32 and A-375. The studies indicate that the substances and their mixtures have an anti-proliferative effect against amelanotic melanoma cells. Additionally, this effect was confirmed in photographic documentation. The preliminary analyses show that a combination of trametinib with inhibitors of autophagy usually causes a greater reduction in the number of living cells than single drugs. The effect is proportional to the concentration of trametinib used. The A-375 cell line is more sensitive to trametinib than the C32 cell line. It was indicated in other studies that CQ and PIK-III (also blocking autophagy), in combination with trametinib, decreased cell viability in A-375 cells [31,32]. A positive effect on the inhibition of autophagy in trametinib therapy has also been demonstrated against pancreatic cancer and lung tumors [33,34]

The cell cycle leads to the growth and development of an organism. Many different factors regulated this process (e.g., checkpoints and kinases). Cancer cells are characterized by cell cycle disorders and uncontrolled proliferation. There are increasing reports indicating an interconnection between the course of the cell cycle and autophagy. Notably, drugs that are cell cycle inhibitors lead to the activation of autophagy, which results in delayed cell death and can lead to resistance to treatment. On the other hand, other antitumor drugs (not anti-cancer genotoxic agents) have been demonstrated to cause overactivated and irreversible autophagy. This action can lead to cell death associated with autophagy [25,35,36,37]. As reported by Lv et al. [38], Li et al. [39], and Thakur et al. [40], chromosome instability mediated by nuclear laminB1 (LMNB1) affects cell autophagy and apoptosis. LMNB1 can ensure the stability of the nuclear structure and influence cellular aging, e.g., by regulating the cell cycle.

Moreover, there is a close correlation between the expression of this protein and the progression and development of melanoma. Cell cycle analysis may be used to evaluate cell population in apoptosis. This process is characterized by the sub-G1 phase and represents the fragmentation of DNA [41]. The studies that were conducted showed that the introduction of autophagy inhibitors into trametinib therapy increases the percentage of cells in the sub-G1 phase. The effect depends on the melanoma cell line. In the C32 cell line, the most increase in the percentage of cells in the sub-G1 phase was observed for 3-MA and 3-MA + TRB samples. In the case of the A-375 cell line, the largest increase was observed for CQ, 3-MA, CQ + TRB, and 3-MA + TRB. It is worth noting that in both populations, treatment with the combination had a better effect than with a single drug. In both lines, there was an accumulation of cells in the G1/G0 phase and a significant decrease in the remaining phases—S and G2/M. In the C32 cells, CQ, TRB, and CQ + TRB caused an increase in the percentage of cells in the G1/G0 phase; in the A-375 cells, only TRB caused an increase in the percentage of cells in the G1/G0 phase. The analyzed changes indicate that the tested substances have an anti-proliferative effect. An important role in the course of cell differentiation, survival, and proliferation is played by the mitogen-activated protein kinase (MEK MAPK/ERK kinase) sailing pathways. Therefore, this signaling cascade has become a target in the search for new cancer therapies. As trametinib is a selective MEK1 and MEK2 kinase inhibitor, its action may affect p44/42 (ERK1/2) levels [42,43]. The obtained results indicate that in the treated C32 cells, there is a significant reduction in the level of p44/42 in all tested samples. The situation is slightly different in the A-375 cell lines; there is a slight decrease in p44/p42 in all tested trials except the sample with trametinib, where there is no decrease, but these changes were not statistically significant. However, these changes may show a disorder of cell proliferation.

Oxidative stress is a condition in which excessive amounts of ROS (reactive oxygen species) appear in cells. This situation can be related to various diseases, e.g., neurodegenerative disease, cardiovascular disease, and cancers. The high ROS level may damage intracellular macromolecules, lipids, and nucleic acids. These damages can lead to a disorder of cell proliferation and, finally, to apoptosis. Low levels of reduced thiols such as glutathione (GSH) in cells may indicate the presence of a large amount of ROS [44,45,46,47]. The conducted studies showed that using treatment caused an increase in the percentage of cells with low GSH levels in all samples, but the effect was different in both cell lines. In the C32 cells, the best effects were for the 3-MA and 3-MA + TRB, but in the A-375 culture, the greatest changes were received for the CQ, 3-MA, CQ + TRB, and 3-MA + TRB; however, the impact of combinations was stronger than that of autophagy inhibitors alone. It indicates that the inhibitor 3-methyladenine mainly induces oxidative stress in cells of the C32 lineage, and in the case of A-375, the enhancement of oxidative stress is due to the potentiation of trametinib with an autophagy inhibitor. According to the data, there is a correlation between oxidative stress and autophagy. The presence of ROS can induce autophagy [48].

Increased production of ROS can result from damage to mitochondria, which are organelles responsible for the process of cellular respiration. About 95% of ROS are derived from the respiratory chain of the inner mitochondrial membrane. Thus, mitochondrial damage can be a source of increased intracellular ROS levels and induce apoptosis [26,48,49,50]. The studies that were conducted showed that mitochondrial membrane abnormalities occurred in all tested trials. In the C32 culture, the percentage of cells with depolarized mitochondria is similar in most samples, while in the A-375 line, the results are much more varied. The highest increase in the percentage of cells with depolarized mitochondria was for CQ, CQ + TRB, and 3-MA + TRB in the A-375 lineage. Thus, the results of the mitochondrial potential analysis correspond to those of reduced thiol level for the A-375 line, suggesting that mitochondrial dysfunction is a major source of ROS in the treatment with the combination of TRB and the autophagy inhibitor of A375 cells.

Apoptosis is a conserved programmed cell death, which is regulated by various factors. Disorders in the normal course of apoptosis may lead to excessive proliferation and development of cancer. Therefore, anti-cancer therapies inducing the process of apoptosis are an important element of effective oncotherapy [51,52,53]. One of the possibilities for determining the process of apoptosis in cells is the annexin V assay. This analysis is based on binding phosphatidylserine because, during apoptosis, a membrane phospholipid is translocated from the inner side of the cell membrane to its outer side [54,55]. The results revealed that the tested agents induce apoptosis, but the effect varies depending on both the culture and the treatment used. Most of the treated cells were in early apoptosis. For the C32 line, the greatest effect on apoptosis induction was observed for 3-MA, which corresponds to the results for the analysis of intracellular reduced thiol level. This suggests that 3-MA may lead to apoptosis through the induction of oxidative stress. The results obtained are consistent with the observations of Chicote et al. [18] and Pickard et al. [19]. Moreover, it is noteworthy that in the case of the C32 line, the combination of chloroquine and trametinib has a much stronger effect than the separate action of these drugs. In the case of the A-375 line, more treated cells are in late apoptosis than in the C32 line. Also, it was observed that combinations of trametinib with autophagy inhibitors exhibit a greater effect than single agents.

One of the most popular markers of autophagy is the microtubule-associated protein LC3 (light chain 3). This protein is responsible for the formation of autophagosomes in mammalian cells. There are three forms of LC3—LC3A, LC3B, and LC3C. LC3B undergoes different post-translation transformations than LC3A and LC3C; this isoform also has distinct functions. The role of LC3B is crucial in apoptosis and differentiation. As data shows, this protein can play an important role in tumorigenesis and resistance to treatment [24,56,57,58]. In the first step, the level of non-selective marker LC3A/B has been marked. In both tested cultures, trials with CQ caused very significant upregulation of this protein. As CQ inhibits autophagy at the autophagosome formation stage, this situation induces an overproduction and cumulation of the LC3 protein [26,59]. Interestingly, trials with 3-MA give various effects depending on the cell line. In the C32 line, there was an increase in LC3 production, in contrast to the A-375 line, where no significant changes in the level of this protein were observed. This differential response of cell lines may be due to the dual role of 3-MA in the modulation of autophagy [26]. The results obtained from the analysis of the level of selective LC3B protein generally confirm the changes detected for LC3A/B. Surprisingly, trametinib, an autophagy activator, when used alone, did not increase the level of LC3.

The presented results indicate the therapeutic potential of combining trametinib with autophagy inhibitors in the treatment of melanoma. The differential response of the studied cell lines may result from the individual characteristics of these lines. Disorders caused by CQ and 3-MA, both alone and in combination, are therapeutically relevant. The research that was conducted shows that the modulation of the autophagy process can contribute to the improvement of the effectiveness of oncological treatment.

## 5. Conclusions

In summary, the obtained results showed that the introduction of autophagy inhibitors into treatment increases the anti-cancer activity of trametinib against amelanotic melanoma cells. The main observed effect in both cells was an anti-proliferative action, which was also evidenced by the p44/42 level; however, it occurred with a different intensity. The C32 cell line is more sensitive to the 3-MA treatment, which was confirmed in the analysis of the cell cycle, the level of reduced thiols, LC3A/B and LC3B, as well as an annexin V assay. For A-375, this line shows sensitivity to CQ and 3-MA, which is confirmed by analysis of the cell cycle, reduced thiol level, mitochondrial membrane potential and annexin V assay. The combination of trametinib with autophagy inhibitors indicated a greater treatment effect than alone therapy in both cell lines. A significant increase in the LC3A/B and LC3B levels in both cultures indicates inhibition of autophagy flux at the stage of autophagosome formation by CQ. The increase in the level of the same proteins in the case of CQ samples is not as significant as in the case of 3-MA (or does not occur at all), which may be due to the inhibition of autophagy at a different point in the molecular pathway.

## Figures and Tables

**Figure 1 cells-14-00557-f001:**
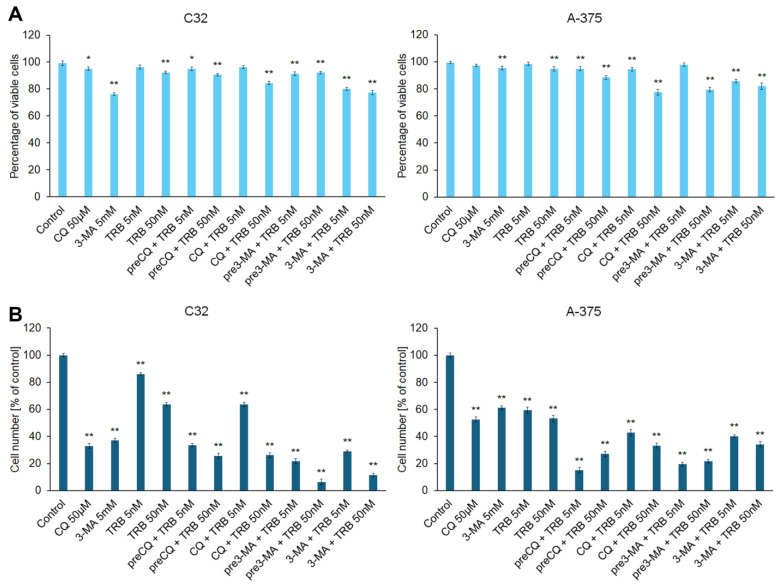
The effects of trametinib (TRB) and autophagy inhibitors: chloroquine (CQ) and 3-methyladenine (3-MA) on the viability (**A**) and proliferation (**B**) of C32 and A-375 melanoma cell lines. The cells were incubated with TRB in concentrations of 5 nM or 50 nM, CQ in a concentration of 50 µM, and 3-MA in a concentration of 5 mM. The agents were used alone (e.g., TRB 50 nM), in co-treatment (e.g., CQ + TRB 50 nM), or in a pre-incubation (pre) model (e.g., preCQ + TRB 50 nM). The bars represent the mean ± SD (standard deviation) of three independent experiments in at least triplicate; * *p* < 0.05, ** *p* < 0.01 vs. control.

**Figure 2 cells-14-00557-f002:**
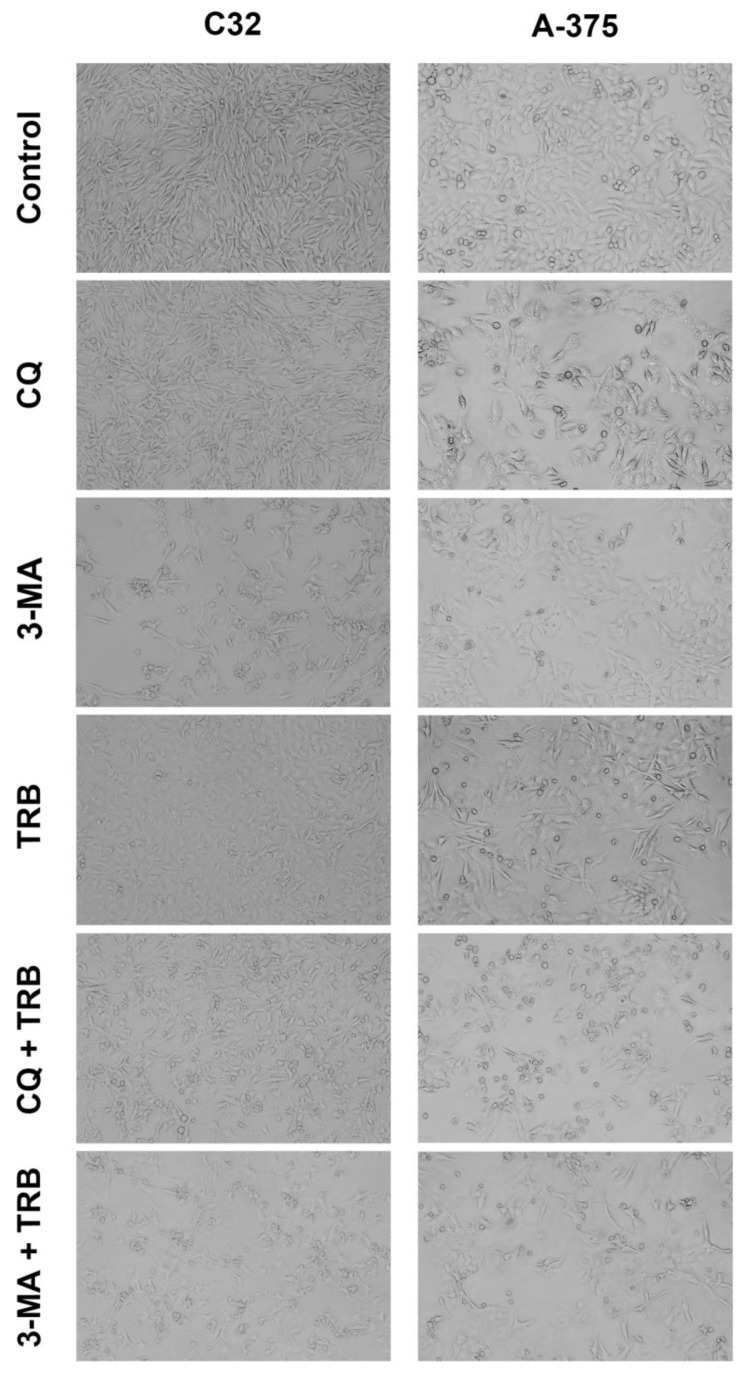
Microscopic analysis of C32 and A-375 cells treated with chloroquine (CQ) in a concentration of 50 µM, 3-methyladenine (3-MA) in a concentration of 5 mM, or/and trametinib (TRB) in a concentration of 50 nM. The cells were observed with an Eclipse TS-100-F microscope.

**Figure 3 cells-14-00557-f003:**
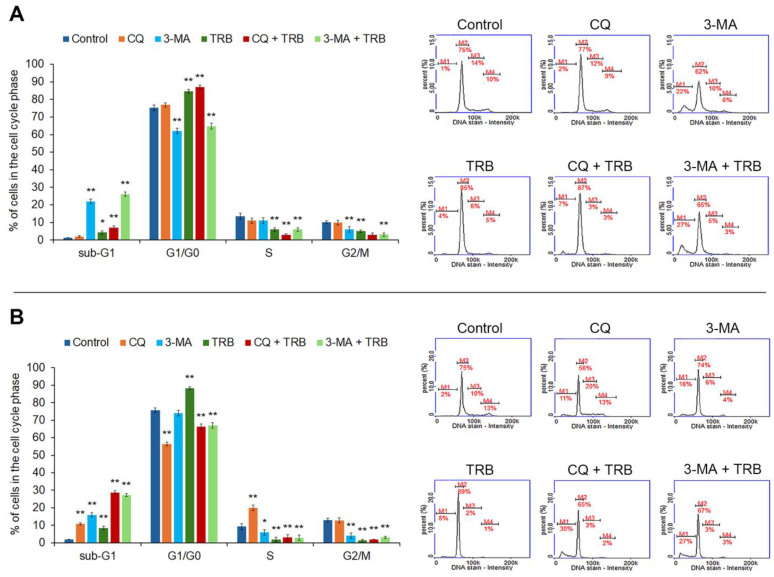
Cytometric cell cycle analysis of C32 (**A**) and A-375 (**B**) cells treated with chloroquine (CQ) in a concentration of 50 µM, 3-methyladenine (3-MA) in a concentration of 5 mM, or/and trametinib (TRB) in a concentration of 50 nM. The left panel bars represent a mean ± SD (standard deviation) of three independent experiments in at least triplicate; * *p* < 0.05, ** *p* < 0.01 vs. control. In the right panel, representative histograms showing the cells in sub-G1(M1), G1/G0 (M2), S (M3) and G2/M (M4) phases are presented.

**Figure 4 cells-14-00557-f004:**
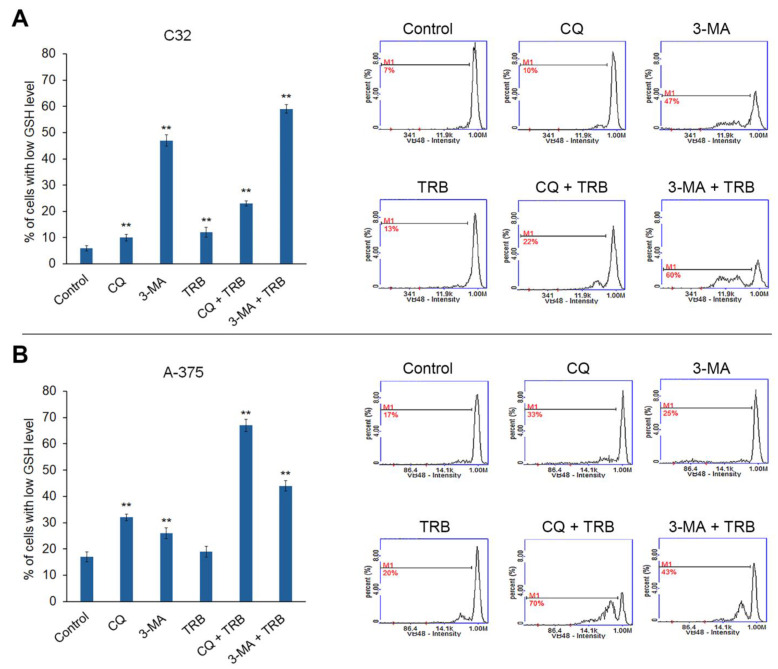
Cytometric analysis of intracellular glutathione in C32 (**A**) and A-375 (**B**) cells treated with chloroquine (CQ) in a concentration of 50 µM, 3-methyladenine (3-MA) in a concentration of 5 mM, or/and trametinib (TRB) in a concentration of 50 nM. The left panel bars represent the mean ± SD (standard deviation) of three independent experiments in at least triplicate; ** *p* < 0.01 vs. control. In the right panel, the representative histograms are presented; M1 = cells with low GSH (reduced glutathione) level.

**Figure 5 cells-14-00557-f005:**
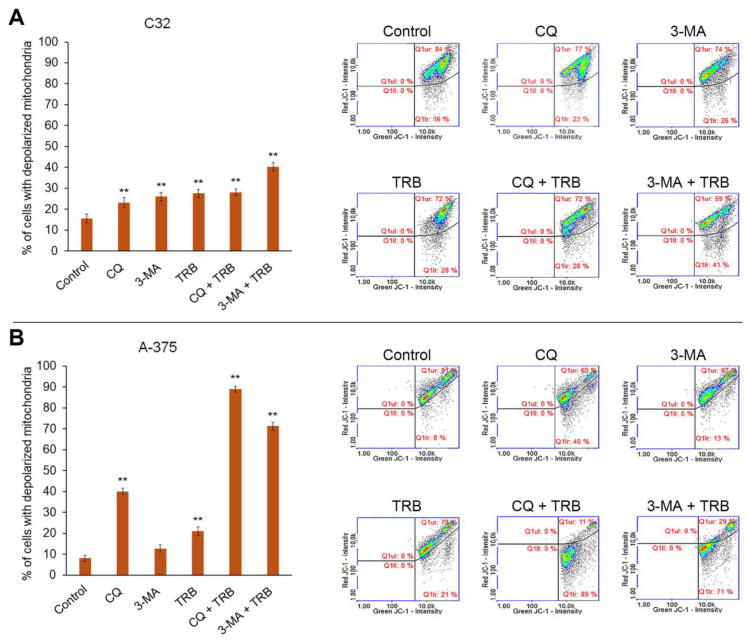
Cytometric analysis of mitochondrial potential in C32 (**A**) and A-375 (**B**) cells treated with chloroquine (CQ) in a concentration of 50 µM, 3-methyladenine (3-MA) in a concentration of 5 mM, or/and trametinib (TRB) in a concentration of 50 nM. The left panel bars represent the mean ± SD (standard deviation) of three independent experiments in at least triplicate; ** *p* < 0.01 vs. control. In the right panel, representative scatter plots are presented. Q1ur—cells with polarized mitochondria; Q1lr—cells with depolarized mitochondria.

**Figure 6 cells-14-00557-f006:**
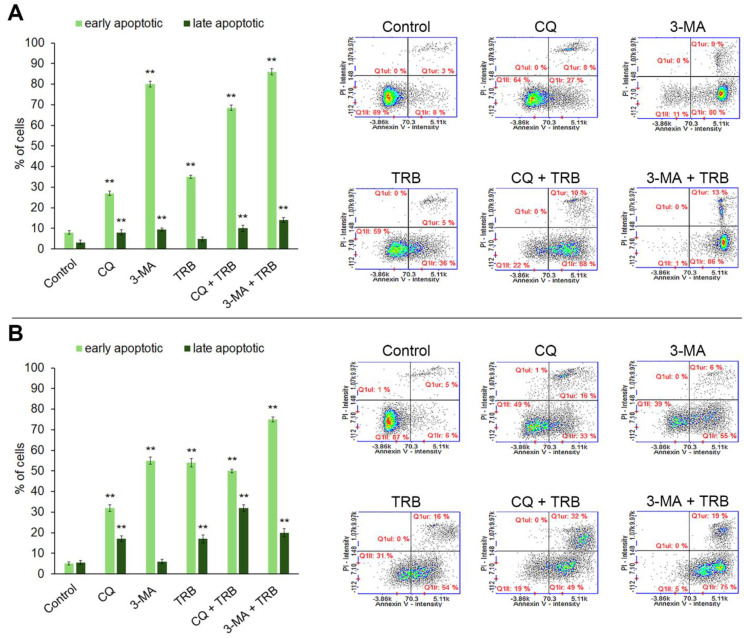
Apoptosis assay with Annexin V and propidium iodide (PI) staining of C32 (**A**) and A-375 (**B**) cells treated with chloroquine (CQ) in a concentration of 50 µM, 3-methyladenine (3-MA) in a concentration of 5 mM, or/and trametinib (TRB) in a concentration of 50 nM. The left panel bars represent the mean ± SD (standard deviation) of three independent experiments in at least triplicate; ** *p* < 0.01 vs. control. In the right panel, representative scatter plots were presented. Q1ll—Annexin V-negative/PI-negative (healthy) cells; Q1lr—Annexin V-positive/PI-negative (early apoptotic) cells; Q1ur—Annexin V-positive/PI-positive (late apoptotic) cells, Q1ul—Annexin V-negative/PI-positive (necrotic) cells.

**Figure 7 cells-14-00557-f007:**
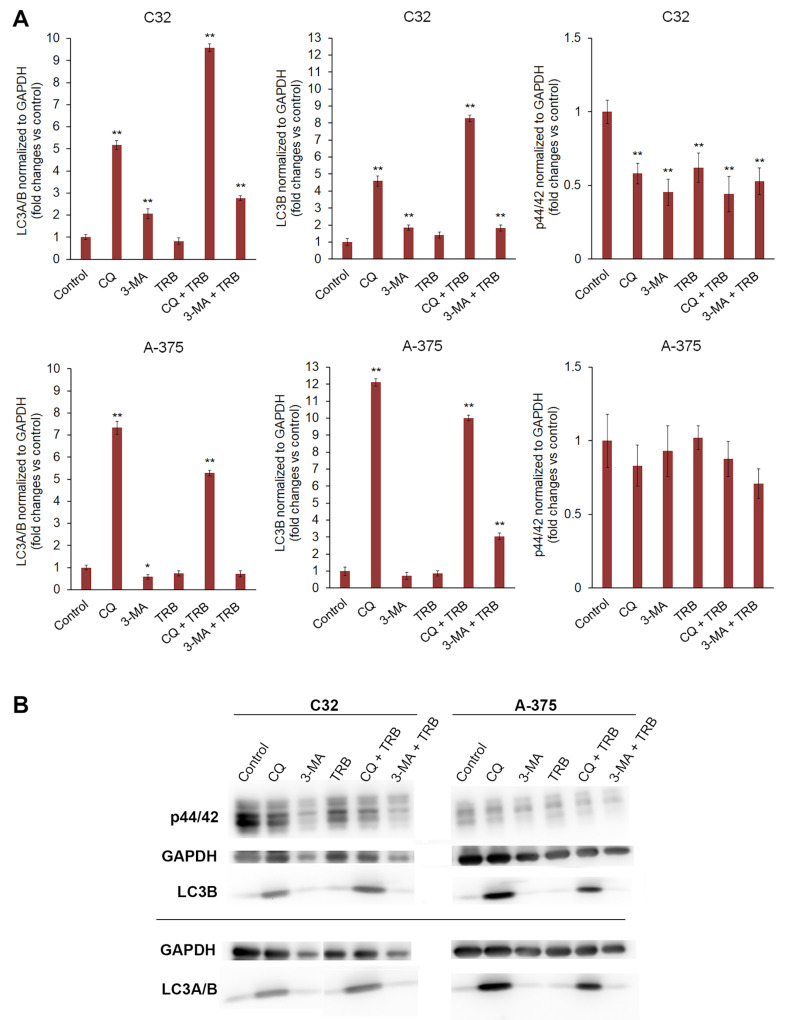
Western blot analysis of p44/42, LC3B, and LC3A/B proteins in C32 and A-375 cell lines treated with chloroquine (CQ) in a concentration of 50 µM, 3-methyladenine (3-MA) in a concentration of 5 mM, or/and trametinib (TRB) in a concentration of 50 nM. In the upper panel bars, (**A**) represents the mean ± SD (standard deviation) of three independent experiments in at least triplicate; * *p* < 0.05, ** *p* < 0.01 vs. control. Corresponding representative blot images are also presented (**B**).

## Data Availability

The data presented in this study are available upon request from the corresponding author.

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
