# Peer review of "The Assessment of the Effect of Autophagy Inhibitors—Chloroquine and 3-Methyladenine on the Antitumor Activity of Trametinib Against Amelanotic Melanoma Cells"

_cells, 2025, doi:10.3390/cells14070557_

Round 1
Reviewer 1 Report
Comments and Suggestions for Authors
This study explores the synergistic effects of combining the MAPK inhibitor trametinib with autophagy inhibitors (chloroquine and 3-methyladenine) in treating amelanotic melanoma. This work underscores the potential of drug combinations targeting both oncogenic signaling and autophagy in aggressive melanoma subtypes.
Comments:
1. The abstract can further highlight the key findings, such as which drug combination is more effective on which cell line. The authors should clearly indicate how the autophagy inhibitors affect the mechanism of action of Trametinib (e.g., whether through regulating specific signaling pathways).
2. It is not clear whether the concentration ranges used in the experiment (e.g., 50 µM Chloroquine and 5 mM 3 - Methyladenine) are based on previous dose optimization. If the concentrations come from references, they should be added.
3. Although the article mentions the use of ANOVA and multiple comparison tests, the specific P-values or effect sizes are not clearly indicated.
4. The labels of some figures (such as Figure 1 and Figure 3) are not clear enough, and readers may have difficulty quickly understanding the meaning of the data.
5. Although the results section presents a large amount of data, the interpretation of the data is rather brief. For example, the authors should add the quantification of cell shape (spreading area, aspect ratio, etc. ) in Fig. 2 to show the change of cell morphology clearly.
6. Chromosome instability mediated by nuclear lamin has recently been shown to impact cell autophagy and apoptosis ( Cell Death Discov. 10, 269 (2024)ï¼› Cancers 16(23) 2024, 4060; PloS one 2021, 16 (7), e0253062). The authors should add this part to the discussion.
Comments on the Quality of English LanguageI recommend asking a native English speaker to check the text.
Author Response
Comment: This study explores the synergistic effects of combining the MAPK inhibitor trametinib with autophagy inhibitors (chloroquine and 3-methyladenine) in treating amelanotic melanoma. This work underscores the potential of drug combinations targeting both oncogenic signaling and autophagy in aggressive melanoma subtypes.
Response: We would like to thank the Reviewer for his valuable comments and suggestions on our manuscript. We have marked the changes in the manuscript file by „track changes” option.
Comment 1: The abstract can further highlight the key findings, such as which drug combination is more effective on which cell line. The authors should clearly indicate how the autophagy inhibitors affect the mechanism of action of Trametinib (e.g., whether through regulating specific signaling pathways).
Response: In accordance with the Reviewer's comment, we have added some information in the summary section, as follows:
Malignant melanoma, particularly an amelanotic melanoma, still contributes to a very serious problem in public health. One way to find new therapies is to learn and understand the molecular pathways that regulate cancer growth and development. In the case of a tumor, the autophagy process can lead to the development or inhibition of cancer. This study aimed to assess the cytotoxicity of connection trametinib (MEK1 and MEK2 kinase inhibitor) with autophagy inhibitors - chloroquine (lysosomal clearance of autophagosomes inhibitor) and 3-methyladenine (phosphatidylinositol 3-kinases inhibitor), on two amelanotic melanoma cell lines (C32 and A-375). The results showed that combination therapy had better anti-proliferative effects than alone therapy in both cell lines. The C32 cell line was more sensitive to 3-methyladenine treatment (alone and in combinations), and the A375 line shows sensitivity to chloroquine and 3-methyladenine (alone and in combinations). The anti-proliferative effect was accompanied by dysregulation of the cell cycle, a decrease in the reduced thiols, the depolarization of the mitochondrial membrane and the level of p44/p42 MAPK. Both inhibitors have the ability to induce apoptosis. Differences in the level of LC3A/B and LC3B proteins between the chloroquine and the 3-methyladenine samples indicate that these drugs inhibit autophagy at different stages. The enhancement of the effect of trametinib by autophagy inhibitors suggests the possibility of combining drugs with anticancer potential with modulators of the autophagy process.
Comment 2: It is not clear whether the concentration ranges used in the experiment (e.g., 50 µM Chloroquine and 5 mM 3 - Methyladenine) are based on previous dose optimization. If the concentrations come from references, they should be added.
Response: The information is included in the results section:
Line 183-184: „The concentrations of chloroquine, 3-methyladenine and trametinib were selected on the basis of literature data [17-23].”
References:
- Shao, T.; Ke, H.; Liu, R.; Xu, L.; Han, S.; Zhang, X.; Dang, Y.; Jiao, X.; Li, W.; Chen, Z.J.; Qin, Y.; Zhao, S. Autophagy regulates differentiation of ovarian granulosa cells through degradation of WT1. Autophagy 2022, 18, 1864-1878. doi: 10.1080/15548627.2021.2005415. Epub 2022 Jan 13. PMID: 35025698; PMCID: PMC9450966.
- Chicote, J.; Yuste, V.J.; Boix, J. and Ribas, J. Cell Death Triggered by the Autophagy Inhibitory Drug 3-Methyladenine in Growing Conditions Proceeds With DNA Damage. Front. Pharmacol. 2020, 11, 580343. doi: 10.3389/fphar.2020.580343
- Pickard, R.D.; Spencer, B.H.; McFarland, A.J.; Bernaitis, N.; Davey, A.K.; Perkins, A.V.; Chess-Williams, R.; McDermott, C.M.; Forbes, A.; Christie, D.; Anoopkumar-Dukie, S. Paradoxical effects of the autophagy inhibitor 3-methyladenine on docet-axel-induced toxicity in PC-3 and LNCaP prostate cancer cells. Naunyn Schmiedebergs Arch. Pharmacol. 2015, 388, 793-9. doi: 10.1007/s00210-015-1104-7. Epub 2015 Feb 24. PMID: 25708950.
- Zhao, F.; Feng, G.; Zhu, J.; Su, Z.; Guo, R.; Liu, J.; Zhang, H.; Zhai, Y. 3-Methyladenine-enhanced susceptibility to sorafenib in hepatocellular carcinoma cells by inhibiting autophagy. Anticancer Drugs 2021, 32, 386-393. doi: 10.1097/CAD.0000000000001032. PMID: 33395067; PMCID: PMC7952045.
- Lee, Y.; Park, D. Effect of Metformin in Combination With Trametinib and Paclitaxel on Cell Survival and Metastasis in Melanoma Cells. Anticancer Res. 2021, 41, 1387-1399. doi: 10.21873/anticanres.14896. PMID: 33788730.
- Yee, P.S.; Zainal, N.S.; Gan, C.P.; Lee, B.K.B.; Mun, K.S.; Abraham, M.T.; Ismail, S.M.; Abdul Rahman, Z.A.; Patel, V.; Cheong, S.C. Synergistic Growth Inhibition by Afatinib and Trametinib in Preclinical Oral Squamous Cell Carcinoma Models. Target Oncol. 2019, 14, 223-235. doi: 10.1007/s11523-019-00626-8. PMID: 30806895.
- Mudianto, T.; Campbell, K.M.; Webb, J.; Zolkind, P.; Skidmore, Z.L.; Riley, R.; Barnell, E.K.; Ozgenc, I.; Giri, T.; Dunn, G.P.; Adkins, D.R.; Griffith, M.; Egloff, A.M.; Griffith, O.L.; Uppaluri, R. Yap1 Mediates Trametinib Resistance in Head and Neck Squamous Cell Carcinomas. Clin. Cancer Res. 2021, 27, 2326-2339. doi: 10.1158/1078-0432.CCR-19-4179. Epub 2021 Feb 5. PMID: 33547198; PMCID: PMC8046740.
Comment 3: Although the article mentions the use of ANOVA and multiple comparison tests, the specific P-values or effect sizes are not clearly indicated.
Response: Thank you for your valuable consideration. We considered adding detailed p-values. However, due to the large amount of data in the graphs, we felt that this addition would make the figures less readable. In the description of the figures, we limited ourselves to a general notation: „*p<0.05, **p<0.01 vs. control”.
Comment 4: The labels of some figures (such as Figure 1 and Figure 3) are not clear enough, and readers may have difficulty quickly understanding the meaning of the data.
Response: Following the Reviewer's comment, the size of all figures (except figure2) has been increased to make the labels more visible. Moreover, in the Figure 1 caption we have added, the example combinations in parentheses as follows:
Figure 1. The effects of trametinib (TRB) and autophagy inhibitors: chloroquine (CQ) and 3-methyladenine (3-MA) on viability (A) and proliferation (B) of C32 and A-375 melanoma cell lines. The cells were incubated with: TRB in concentrations of 5 nM or 50 nM, CQ in a concentration of 50 µM, 3-MA in a concentration of 5 mM. The agents were used alone (e.g. TRB 50nM), in co-treatment (e.g. CQ+TRB 50nM) or in a pre-incubation (pre) model (e.g. preCQ+TRB 50nM). Bars represent mean ± SD (standard deviation) of 3 independent experiments in at least triplicate; p*<0.05, **p<0.01 vs. control
Comment 5: Although the results section presents a large amount of data, the interpretation of the data is rather brief. For example, the authors should add the quantification of cell shape (spreading area, aspect ratio, etc. ) in Fig. 2 to show the change of cell morphology clearly.
Response: Thank you for your attention. The microscopic analyses were aimed at the overall evaluation of cell morphology. The observations showed that the effect was observed in both the C32 and A-375 cell lines. For the C32 line, the strongest antiproliferative effect was seen with 3-methyladenine, particularly when combined with trametinib. Additionally, the cells in this sample exhibited an expanded, non-spherical morphology. In the case of the A-375 line, chloroquine showed the most significant effect, especially when combined with trametinib. These cells tended to adopt a rounded shape. Notably, in samples treated with trametinib, the A-375 cell line also displayed an expanded, non-spherical morphology.
Comment 6: Chromosome instability mediated by nuclear lamin has recently been shown to impact cell autophagy and apoptosis ( Cell Death Discov. 10, 269 (2024)ï¼› Cancers 16(23) 2024, 4060; PloS one 2021, 16 (7), e0253062). The authors should add this part to the discussion.
Response: Thank you for your valuable comment, we have included according to this:
The cell cycle leads to the growth and development of an organism. Many different factors regulated this process (e.g. checkpoints, kinases). Cancer cells are characterized by cell cycle disorders and uncontrolled proliferation. There are more and more reports indicating an interconnection between the course of the cell cycle and autophagy. Notably, drugs that are cell cycle inhibitors lead to the activation of autophagy, which results in delayed cell death and can lead to resistance to treatment. On the other hand, other antitumor drugs (not anticancer genotoxic agents) have been demonstrated to cause overactivated and irreversible autophagy. This action can lead to cell death associated with autophagy [25,36,37]. As reported by Lv et al. [38], and Li et al. [39] and Thakur et al. [40] chromosome instability mediated by nuclear laminB1 (LMNB1) affects cell autophagy and apoptosis. LMNB1 can ensure the stability of the nuclear structure and influence cellular aging, e.g. by regulating the cell cycle. Moreover, there is a close correlation between the expression of this protein and the progression and development of melanoma. Cell cycle analysis may be used to evaluate cell population in apoptosis. This process is characterized by sub-G1 phase and represents the fragmentation of DNA [38 40 41]. The conducted studies showed that the introduction of autophagy inhibitors into trametinib therapy increases the percentage of cells in the sub-G1 phase. The effect depends on the melanoma cell line. In the C32 cell line the most increase in the percentage of cells in the sub-G1 phase was observed for 3-MA and 3-MA+TRB samples. In the case of the A-375 cell line, the most increase was observed for CQ, 3-MA, CQ+TRB and 3-MA+TRB. It is worth noting that in both populations, treatment with the combination had a better effect than it with a single drug. In both lines, there was an accumulation of cells in G1/G0 phase and significant decrease in the remaining phases - S and G2/M. While in C32 cells, CQ, TRB and CQ+TRB caused an increase in the percentage of cells in G1/G0 phase, in A-375 cells, only TRB caused an increase in the percentage of cells in G1/G0 phase. The analyzed changes indicate that the tested substances have an anti-proliferative effect. An important role in the course of cell differentiation, survival and proliferation is played by the mitogen-activated protein kinase (MEK MAPK/ERK kinase) sailing pathways. Therefore, this signalling cascade has become a target in the search for new cancer therapies. As trametinib is a selective MEK1 and MEK2 kinase inhibitor, its action may affect p44/42 (ERK1/2) levels [39 41 42,40 42 43]. The obtained results indicate that in the treated C32 cells, there is a significant reduction in the level of p44/42 in all tested samples. The situation is slightly different in the A-375 cell lines, there is a slight decrease in p44/p42 in all tested trials except the sample with trametinib, where there is no decrease, but these changes were not statistically significant. However, these changes may show a disorder of cell proliferation.
Reviewer 2 Report
Comments and Suggestions for Authors
The incidence of melanoma is increasing worldwide. Trametinib has been used to treat melanoma. However, its use faces some difficulties such as side effects. The present study aimed to assess the effects of two autophagy inhibitors, chloroquine and 3-methyladenine in combination with trametinib, on two amelanotic melanoma cell lines, C32 and A-375. The study examined two autophagy inhibitors against two cell lines showing that trametinib and the two autophagy inhibitors 1) are cytotoxic to the melanoma cells, 2) disrupted cell cycle of melanoma cells, 3) decreased the level of reduced thiols, 4) decreased mitochondrial membrane potential, 5) induced apoptosis, and 6) changed the levels of LC3A etc. These results are extensive and informative, which warrants its publication. The reviewer have two minor comments as follow.
Minor comments:
- Please mention at the end of Introduction as to why the authors selected chloroquine and 3-methyladenosine as autophagy inhibitors.
- Please speculate what would happen if chloroquine and 3-methyladenosine are examined against melanotic melanoma? This is important when one considers the well-known affinity of chloroquine against melanin.
Author Response
Comment: The incidence of melanoma is increasing worldwide. Trametinib has been used to treat melanoma. However, its use faces some difficulties such as side effects. The present study aimed to assess the effects of two autophagy inhibitors, chloroquine and 3-methyladenine in combination with trametinib, on two amelanotic melanoma cell lines, C32 and A-375. The study examined two autophagy inhibitors against two cell lines showing that trametinib and the two autophagy inhibitors 1) are cytotoxic to the melanoma cells, 2) disrupted cell cycle of melanoma cells, 3) decreased the level of reduced thiols, 4) decreased mitochondrial membrane potential, 5) induced apoptosis, and 6) changed the levels of LC3A etc. These results are extensive and informative, which warrants its publication. The reviewer have two minor comments as follow.
Reponse: We would like to thank the Reviewer for his valuable comments and suggestions on our manuscript. We have marked the changes in the manuscript file by „track changes” option
Comment 1: Please mention at the end of Introduction as to why the authors selected chloroquine and 3-methyladenosine as autophagy inhibitors.
Response: In accordance with the Reviewer's comment, we have added some information in the introduction section, as follows:
The aim of the study was to assess the effect of autophagy inhibitors on the antitumor activity of trametinib against amelanotic melanoma cells. An important aspect of the study was the inhibition of autophagy at different stages of the process. Therefore, chloroquine inhibited this process at the autophagosome completion phase, and 3-methyladenine at the nucleation phase. Understanding the mechanism of action and the impact of autophagy on the development of cancer may contribute to improving the quality and effective-ness of current cancer treatment. The introduction of an autophagy inhibitor into anti-cancer therapy may contribute to overcoming drug resistance and increasing the cytotoxic effect.
Comment 2: Please speculate what would happen if chloroquine and 3-methyladenosine are examined against melanotic melanoma? This is important when one considers the well-known affinity of chloroquine against melanin.
Response: Thank you for your valuable consideration. The subject on the use of autophagy inhibitors on different types of melanoma is very interesting, not only because of the modulation of the process itself, but also considering the affinity to melanin of the drugs used (chloroquine). We are going to conduct studies on melanotic melanoma lines and publish them in the future.